# Nature-Based Interventions for Psychological Wellbeing in Long-Term Conditions: A Systematic Review

**DOI:** 10.3390/ijerph19063214

**Published:** 2022-03-09

**Authors:** Eleanor M. Taylor, Noelle Robertson, Courtney J. Lightfoot, Alice C. Smith, Ceri R. Jones

**Affiliations:** 1Department of Neuroscience, Psychology and Behaviour, University of Leicester, Leicester LE1 7HA, UK; nr6@leicester.ac.uk; 2Leicester Kidney Lifestyle Team, Department of Health Sciences, University of Leicester, Leicester LE1 7RH, UK; courtney.lightfoot@leicester.ac.uk (C.J.L.); alice.smith@leicester.ac.uk (A.C.S.)

**Keywords:** long-term conditions, nature-based intervention, systematic review, nature, physical health

## Abstract

Background: With the global burden of disease increasing, particularly in relation to often preventable chronic diseases, researchers and clinicians are keen to identify interventions that can mitigate ill health and enhance the psychological wellbeing of people living with long-term conditions (LTCs). It is long established that engagement with nature can support human health and wellbeing, and in recent years, nature-based interventions (NBIs) have been advanced as of potential benefit. This review thus sought to systematically appraise published evidence of the application of NBIs to address psychological wellbeing for those living with LTCs. Methods: A systematic search of three databases, PsycINFO, MEDLINE and SCOPUS, was undertaken, and the BestBETs quality assessment checklist was used to appraise methodological quality of elicited studies. Results: Of 913 studies identified, 13 studies (12 using quantitative methods, one qualitative) were used. Included papers reported use of a variety of psychological outcomes alongside more circumscribed physiological outcomes. Quality appraisal showed modest robustness, some methodological weaknesses and a dominance of application in developed countries, yet synthesis of studies suggested that reported psychological and physiological outcomes present a strong argument for NBIs having a promising and positive impact on psychological wellbeing. Conclusions: NBIs have positive psychological and physiological impacts on people with LTCs, suggesting they may be a suitable addition to current maintenance treatment. Future research should focus on minimising study bias and increasing the potential for cross-cultural applications.

## 1. Introduction

Long-term conditions (LTCs) are understood as diseases with no current cure, yet they require management with drugs or other treatments [1]. Within the UK, those LTCs most associated with premature death comprise diagnoses of cancer, cardiovascular disease (CVD), stroke, lung disease and liver disease [2]. Type II diabetes mellitus (T2DM) also affects 90% of the 3.9 million people currently diagnosed with diabetes in the UK [3] and is linked to the risk for developing other LTCs [2]. Indeed, uncontrolled diabetes and high blood pressure are the biggest known causes of chronic kidney disease (CKD) [4], which affects approximately three million people in the UK. Lifestyle factors, such as poor diet, physical inactivity, drinking alcohol and smoking, can impact disease morbidity and progression [2]. These behaviours increase the risk of developing many LTCs and are modifiable.

Given the enduring and costly nature of LTCs, with in excess of 15 million people in England living with LTCs [1,2], there is a need to improve treatment options. For example, data from the last decade suggest that LTCs account for 50% of GP appointments, 64% of outpatient appointments and 70% of inpatient bed days in England alone [5]. LTCs also affect individuals disproportionately; for example, 58% of those living with LTCs are over 60 years old, while the chance of developing an LTC is 60% more likely and will be 30% more severe for those in the lowest socio-economic groups compared to those in the highest [5].

In an increasingly urbanised world, with a trend for population movement to cities, there is a suggestion that engagement with and exposure to nature is beneficial. For example, those living in areas close to greenspace have a reduced mortality risk [6,7,8]. Greenspace proximity is also associated with a reduced incidence of neurological disorders [9], CVD [7] and T2DM [10]. A recent review of systematic reviews of the relationship between public health and proximity to nature concluded that being closer to greenspace reduces the incidence of stroke, hypertension, dyslipidaemia, asthma and coronary heart disease [11]. Even bringing nature inside has shown advantages for health and wellbeing; for example, Ulrich’s seminal study [12] demonstrated the benefit of views of nature for post-surgical recovery.

The advantageous effects of natural world experiences on human health and wellbeing are increasingly well-established [13]; however, the mechanisms affecting these improvements remain speculative [14]. A dominant putative mechanism is advanced through the biophilia hypothesis, which suggests that humans are innately attracted to natural environments [15], and evolutionary psychology proposes that the human brain and body have been shaped by millions of years living in nature [16]. This intrinsic appeal has fuelled interest in and the development of nature-based interventions (NBIs) to improve health and wellbeing.

NBIs are a diverse array of activities or programmes aimed at engaging individuals in nature-based experiences to improve health and wellbeing. Common NBIs include horticultural therapy, which has been shown to improve general wellbeing [17], as well as mood and performance for people with mental health conditions [18]. Forest therapy is also understood as a method of NBI, most notably in the form of “forest bathing” or *Shinrin Yoku* (reflecting its origins). This involves experiencing the calm and quiet of trees for relaxation with significant benefits for both physical and mental health [19] and human immune function [20], an effect not replicated in an urban comparison [21], as well as improved self-efficacy, life satisfaction and physical activity and reduced unhealthy eating in young people [22]. The beneficial effects of NBIs appear not just attributable to increased physical activity since even short-term visits to urban greenspaces can reduce blood pressure and heart rate variability in comparison to visiting urban streets [23]. In addition to these physiological benefits, there are also psychological benefits of interaction with nature. For example, cognition can also be enhanced, with studies reporting improved Stroop test performance from walking in natural as opposed to built-up environments [24], as well as improved cognitive performance in individuals with major depressive disorder [25].

Incorporated within NBIs are also those which feature human–animal interactions in outdoor environments, involving specially trained animals and therapeutic goals. Previous reviews have revealed the positive effect of animal-assisted interventions for physical and psychological wellbeing [26,27,28,29], as well as the ability to cope with stress, improvements to cardiovascular health, and maintaining health and mobility in older age [30]. More specifically, in addition to mechanical benefits [31,32] such as improved motor ability, independence of ambulation and gait [33], enhanced quality of life (QoL) has been reported following horse therapy for stroke survivors [34]. Horse therapy also appears to confer psychological benefits for those living with chronic back pain who report increases in positive affect and meaningful activities [35], as well as facilitating positive self-identity in people with physical disabilities [36].

Given that a preliminary scoping review revealed no previous overarching review of published evidence examining how NBIs have been offered for those living with LTCs, our review seeks to systematically appraise, synthesise and evaluate published research examining the impact of NBIs on psychological wellbeing. This hopes to identify the extent of work to date which has deployed NBIs and with what effects, assess nature’s impact on wellbeing and its potential to benefit self-management and contribute to the menu of interventions addressing health outcomes, particularly of psychological wellbeing.

## 2. Materials and Methods

### 2.1. Study Selection

This review focused on the most prevalent LTCs associated with premature death [2] but excluded cancers (thus CVD, stroke, lung disease and liver disease). The review also included CKD and T2DM, given their population prevalence. Cancer studies were excluded from the present review because initial scoping identified sufficient number of papers to warrant a separate review. Additionally, given cancers are not exclusively caused by lifestyle factors, nor are they always incurable, they are arguably outside of the focus of this review. This review considered NBIs if the interventions contained a nature or green element, whether conducted inside or outside. Animal-based interventions conducted inside, for example, pet-assisted therapy conducted at a hospital bedside, were not included in this review as they did not meet the inclusion criteria of containing a nature or green element. However, those delivered within an outdoor natural environment were included.

### 2.2. Eligibility Criteria

#### 2.2.1. Inclusion Criteria

Nature-based intervention conducted inside or outside or animal-based intervention conducted in outdoors environments;Involves an active intervention rather than just passive proximity to, e.g., greenspace;Studies with a minimum of 50% participants with at least one diagnosed long-term physical health condition, limited to cardiovascular disease, stroke, lung and liver disease, type II diabetes or chronic kidney disease, but not excluding other co-morbidities;Measuring psychometrically robust psychological outcomes, including (but not limited to) QoL.

#### 2.2.2. Exclusion Criteria

Not specific to long-term conditions listed above;Participants within end-of-life pathways (e.g., palliative care).

### 2.3. Search Strategy

The search strategy was based on the PICO framework [37] (Table 1). A title, abstract and keyword search (see Table 2 for terms) was performed on the following electronic databases in January 2022: PsycINFO, MEDLINE and Scopus. Search limitations included English language and human participants, but the search was not limited to peer-reviewed publications only. After duplicate removal, a manual search of the titles and abstracts was performed using the eligibility criteria listed above.

### 2.4. Data Extraction

Shortlisted articles were read in full by the first author, and a data extraction table adapted from Brooks et al. [38] was used to synthesise data relevant to this review (see Table 3 for a summary). The 12 full-text articles that were excluded prior to analysis were deemed not appropriate for inclusion. Reasons for exclusions included: not having a condition that matched the inclusion criteria for an LTC (*n* = 5), not containing a nature or green element (*n* = 4), protocol-only papers and did not contain any data (*n* = 2) or deemed poor quality as it did not report any data (*n* = 1).

Variables reported were publication year, study design, sample size and participant demographics, type of LTC, type of NBI, psychological and physiological measures used and their outcomes, and country of study. Key information on each study’s main aims were also extracted where they were relevant to the review question. Papers were then grouped by the LTC of participants. Eleven out of thirteen studies included in this review also investigated physiological measures in addition to psychological measures. These are also included in the analysis because understanding is still evolving regarding the bidirectional impacts of psychological and physiological processes.

### 2.5. Quality Appraisal

The BestBETs quality assessment checklist [39], chosen from the Systematic Review Toolbox [40] as recommended by Booth et al. [41], was used to assess the methodological quality of the 12 quantitative studies elicited. This tool provides critical appraisal for a range of study types, which was suited to the diverse study designs in the present review and is designed for cohort healthcare research. It comprises 33 items evaluating domains of objectives and hypotheses, design, measurement and observation, presentation of results, analysis, discussion, interpretation and implementation. For each item, studies were scored 0 (not reported), 1 (reported but inadequate) or 2 (reported adequately), with a potential total score of 64 (see Appendix A). The one qualitative study was assessed using a guide for reading qualitative studies presented by Sandelowski and Barroso [42], which encourages readers of qualitative research to take meaning from the text over employing strict standards and criteria.

## 3. Results

The database search identified 913 studies. Following duplicate removal, a manual search of title and abstracts identified 25 studies, which received a full-paper review. The search also identified a systematic review of systematic reviews [43], which was manually searched and identified a further two studies. In total, 13 studies were eligible for inclusion: twelve quantitative and one qualitative (see PRISMA flow diagram, Figure 1; [44]).

### 3.1. Study Characteristics

The 13 studies identified by this review were published between 2005 and 2020 and reported on a total of 512 participants (see Table 3 for a summary of study characteristics). Eight studies recruited both males and females [34,45,46,47,48,49,50,51], four recruited only males [52,53,54,55] and one did not specify sex [56]. Ten studies were conducted in Asia (including Korea, China or Japan), and one each within the USA, Sweden and Brazil. Most of the studies did not specify how they recruited their participants, although the majority appeared to use opportunistic sampling, while three explicitly stated advertisement using either posters [45], newspapers [53] or within local health centres [48,54].

**Table 3 ijerph-19-03214-t003:** Summary of study characteristics.

Author (Year)	Research Design	Participants	Intervention	Control	Psychological Measurements	Psychological Outcomes	Physiological Measurements	Physiological Outcomes
Beinotti et al. (2013) [34]	Single-blind RCT	20 patients (6 female) ≥1 year post-stroke Mean age 55.5 years	16 weeks physio and horse-riding therapy (10 participants) Mean age 59 years	16 weeks physio only (10 participants) Mean age 52 years	Medical Outcomes Study 36-item Short-Form health survey Measurements before and after the intervention	Significant improvement in functional capacity, physical aspects and mental health following horseback riding therapy compared to controls. No changes in general health state, vitality or emotional aspects	N/A	N/A
Chun et al. (2017) [45]	Two-sample randomised cohort	59 participants (19 female) Mean age 60.8 years (SD 9.1)	Forest therapy programme (30 participants) 4-day trip involving meditation, experiencing the forest and walking	Urban comparison (29 participants) 4-day trip involving meditation and walking in the hotel	Beck Depression Inventory (BDI) Hamilton Depression Rating Scale (HAM-D17) Spielberger State-Trait Anxiety Inventory (STAI) Measurements before and after the intervention	Reduced BDI, HAM-D17 and STAI scores following forest therapy programme compared to baseline. Increased STAI scores in urban group following programme	Oxidative stress: total oxidant capacity and iron-reducing activity Measurements before and after the intervention	No significant differences between forest and urban groups
Pohl et al. (2018) [46]	Qualitative exploration	18 participants (6 female) Mean age 60.3 1–5 years post-stroke	12-week multi-modal intervention incorporating horseback riding	No comparison group	Individual face-to-face semi-structured interviews	Four themes identified: transformative experiences, human–horse interaction, togetherness and belonging, and the all-in-one solution	N/A	N/A
Jia et al. (2016) [47]	Two-sample randomised cohort	20 COPD patients (6 female) Mean age 70.1 years	Forest walking (10 participants) Mean age 70.1 (range 67–77) 7 days at forest site with scheduled walking time, staying in a hotel	City walking (8 participants) Mean age 70 (range 61–79) 7 days at city site with scheduled walking time, staying in a hotel	Profile of Mood State (POMS) Measurements before and after the intervention	Lower POMS scores of “tension–anxiety”, “depression” and “anger–hostility” in forest but not city group	Lymphocytes: NK, NKT-like and CD8+ T-cells expression of intracellular perforin and granzyme B Pro-inflammatory cytokines: interferon-γ (IFN-γ), interleukin-6 (IL-6), interleukin-8 (IL-8), interleukin-1β (IL-1β), tumour necrosis factorα (TNF-α) and C-reactive protein (CRP) COPD-associated factors: pulmonary and activation-regulated chemokine (PARC/CCL-18), surfactant protein D (SP-D) and tissue inhibitor of metalloproteinase (TIMP-1) Stress hormones: serum cortisol and epinephrine Measurements before and after the intervention	Lymphocytes: no significant group difference in proportion of NK, NKT-like and CD8+ T-cells, nor their expression of granzyme B. Significant reduction of NK, NKT-like and CD8+ T-cell expression of intracellular perforin after forest bathing but not city group. Pro-inflammatory cytokines: significant reduction of IFN-γ, IL-6 and IL-8 after forest bathing but not city group. Slight decrease in IL-1β, TNF-α and CRP after forest bathing but not city group. COPD-associated factors: significant decrease in PARC/CCL-18 TIMP-1 after forest bathing but not city group. No significant change in SP-D in either group. Stress hormones: significant decrease in serum cortisol and epinephrine after forest bathing but not city group
Song et al. (2015) [52]	Two-sample randomised cross-over cohort study	20 male participants with high-normal blood pressure (HNBP) or hypertension Mean age 58.0 years (SD 10.6)	Forest walking 17 min walk All participants completed both interventions on 2 consecutive days (10 in each group, counterbalanced: forest first vs urban first)	Urban walking 17 min walk	Semantic Differential (SD) method Profile of Mood State (POMS) Measurements taken at end of each walk	Increased SD scores of “comfortable”, “relaxed” and “natural” after waking in forest area compared with urban area. Reduced negative POMS scores of “tension–anxiety”, “depression”, “anger–hostility”, “fatigue” and “confusion”, with increased “vigour” after walking in forest area compared to urban area	Heart rate variability (HRV) and heart rate Measures collected at 1 min intervals and averaged over the 17 min course	Significantly higher parasympathetic activity during forest walking compared to urban walking. No significant difference in sympathetic nerve activity between groups. Significantly lower mean heart rate during forest walking compared to urban walking. Physiological measures were significantly related to the differences in air temperature and humidity between the forest and urban environments
Li et al. (2016) [53]	Single-sample cross-over cohort study	19 male participants with high-normal blood pressure (HNBP) or hypertension Mean age 51.2 years (SD 8.8)	Forest walking Day trip All participants completed both interventions. Urban first	Urban walking Day trip	Profile of Mood State (POMS) Measurements taken before, during and after each intervention	Reduced POMS (D), (A), (F), (C) and increased (V) in forest walking but not city walking. City group also had increased (D)	Blood pressure and heart rate. Serum triglycerides, total cholesterol (Cho), low-density lipoprotein (LDL) Cho, high-density lipoprotein (HDL) Cho and remnant-like particles (RLP) Cho, serum adiponectin, blood glucose, serum insulin, serum dehydroepiandrosterone sulphate (DHEA-S), serum high-sensitivity C-reactive protein (hs-CRP). Urinary adrenaline, noradrenaline and dopamine (corrected for creatinine) Blood and urine collected in the morning before and after each day trip. Blood pressure and heart rate measured by an ambulatory monitor every 20 min	No significant difference in blood pressure between forest and urban day trips. Significant decrease in heart rate during forest walking compared to urban walking. No significant change in serum triglycerides, Cho, LDL Cho, HDL Cho, and RLP Cho, blood glucose, serum insulin, serum DHEA-S, or hs-CRP. Significant increase in serum adiponectin after forest but not urban day trips. Both forest and urban walking significantly reduced urinary noradrenaline. Non-significant decrease in urinary adrenaline after forest walking compared to urban walking. Significant decrease in urinary dopamine after forest walking compared to urban walking
Ochiai et al. (2015) [54]	Single-sample cohort	9 male participants with high-normal blood pressure (HNBP) Mean age 56.0 years (SD 13.0)	Forest therapy 1-day programme involving walking, sitting and lying down	No comparison group	Semantic Differential (SD) method Profile of Mood State (POMS) combined POMS Total Mood Disturbance (TDM) Measurements before and after the intervention	Increased SD scores of “relaxed” and “natural” after forest therapy compared with baseline. Reduced negative POMS scores of “tension–anxiety”, “confusion” and “anger–hostility”, and TDM after forest therapy.	Blood pressure Urinary adrenaline (corrected for creatinine) Serum cortisol Blood pressure collected during intervention using portable device. Urine and blood samples collected in the afternoon before and after the intervention	Significant decrease in blood pressure after forest therapy. Significant decrease in urinary adrenaline and serum cortisol after forest therapy
Song et al. (2017) [55]	Two-sample, randomised cross-over cohort	20 males with high-normal blood pressure or hypertension Mean age 58.0 years (SD 10.6)	Viewing forest landscape for 10 min while sitting (10 participants saw forest first, 10 saw urban first on 2 consecutive days)	Viewing urban landscape for 10 min while sitting	Modified semantic differential (SD) method completed after each viewing	Significantly increased scores of “comfortable”, “relaxed” and “natural” after viewing forest area compared to urban area	Heart rate variability (HRV) and heart rate collected at 1 min intervals and averaged across the 10 min period	Significantly increased high-frequency HRV during forest compared to urban viewing. No significant difference between high-frequency/low-frequency heart rate significantly lower during forest compared to urban viewing
Sung et al. (2012) [48]	Non-randomised controlled trial	56 participants (22 female) with hypertension Mean age 64.5 years	3-day CBT Forest Therapy programme including 2 recreational visits to forest sites (28 participants) Mean age 63 years (SD 11) 50% male	Provided with printed educational materials for hypertension management (28 participants) Mean age 66 years (SD 7) 28% male	QoL with 5 domains: General Health (GH), Physical Dimension (PD), Mental Dimension (MD), Social Dimension (SD and Hypertension-related Dimension (HTD). Measured at initial visit and at 8-week final visit	Forest group showed significantly increased total QoL scores after forest therapy. Increases in MD and HTD but not GH or SD. No significant change in control group	Blood pressure: measured at start and at end of 3-day program. Daily self-monitoring morning and evening from first until last day of study Salivary cortisol: collected at initial visit and 8-week final visit	Blood pressure: marginally significantly larger decrease in systolic blood pressure following forest therapy (at day 3). No change in diastolic blood pressure or either of self-measured at week 4 or 8. No significant longitudinal change in blood pressure in either group Salivary cortisol: significantly larger reduction following forest therapy and significant increase in control group
Wu et al. (2020) [49]	Two-sample randomised cohort	31 participants (12 female) with hypertension Mean age 73.7 years	Forest walking (20 participants) 2 days at forest site with scheduled walking, rest and staying in a hotel	City walking (11 participants) 2 days at city site with scheduled walking, rest and staying in a hotel	Profile of Mood State (POMS) Measurements before and after the intervention	Reduced negative POMS scores of “tension–anxiety”, “depression”, “confusion” and “fatigue”, as well as increased “vigour” after forest bathing compared to city walking group.	Blood pressure, heart rate, oxygen saturation (SpO2%) and heart rate variability (HRV) Measurements taken in the morning before and after the intervention	Significant decrease in diastolic blood pressure, but not systolic blood pressure after forest bathing compared to controls Significant increase in SpO2% after forest bathing compared to controls No significant change in heart rate Significantly decreased in LF HRV and LF/HF HRV after forest bathing compared to controls. Significant increase in HF HRV after forest bathing compared to controls
Mao et al. (2012) [56]	Two-sample randomised cohort	24 patients with essential hypertension (does not specify sex or age)	Forest walking (12 participants) 7 days at forest site with scheduled walking time, staying in a hotel	City walking (12 participants) 7 days at city site with scheduled walking time, staying in a hotel	Profile of Mood State (POMS) Measurements before and after the intervention.	Lower POMS scores of “depression”, “anger-hostility”, “fatigue” and “confusion”, with increased “vigour” in forest but not city group	Blood pressure and heart rate Cytokines: homocysteine (Hcy), constituents of the renin-angiotensin system (RAS) including renin, angiotensinogen (AGT), angiotensin II (Ang II), angiotensin II type 1 receptor (AT1), angiotensin II type 2 receptor (AT2) Cardiovascular disease-associated factors: serum interleukin-6 (IL-6), tumour necrosis factorα (TNF-α) and endothelin-1 (ET-1) Measurements taken in the morning before and after the intervention	Significant decrease in systolic and diastolic blood pressure after forest bathing compared to controls No significant change in heart rate Significant decrease in ET-1 and Hcy, RAS constituents including AGT, AT1, and AT2 after forest bathing compared to controls. Non-significant reduction in renin and Ang II after forest bathing compared to controls. Significant association between systolic blood pressure and Ang II, ET-1 and Hcy. Diastolic blood pressure was significantly associated with Ang II and ET-1. BP was poorly associated with the change in renin, AT1, and AGT Significant decrease in serum IL-6 after forest bathing compared to controlsNo significant change in TNF-α
Mao et al. (2017) [50]	Two-sample randomised cohort	33 participants (14 female) with Chronic Heart Failure Mean age 71.8 years	Forest walking 23 participants 4-day trip	City walking 10 participants 4-day trip	Profile of Mood State (POMS) Measurements before and after the intervention.	Reduced negative POMS scores of “tension–anxiety”, “depression”, “anger–hostility” and “confusion” compared to baseline in forest group but not city group.	High-sensitive-reactive protein (hCRP) Bio-markers for heart failure: BNP and NT-ProBNP Cardiovascular disease-related factors: ET-1, constituents of the renin–angiotensin system (RAS) including renin, angiotensinogen (AGT), angiotensin II (Ang II), angiotensin II type 1 receptor (AT1), angiotensin II type 2 receptor (AT2) Pro-inflammatory cytokines: interleukin-6 (IL-6), tumour necrosis factorα (TNF-α) Oxidative indicators: activity for serum total SOD (T-SOD) and lipid peroxidation reflected by malondialdehyde (MDA) Measurements taken in the morning before and after the intervention	Significant decrease in BNP after forest bathing compared to baseline. No significant difference in controls. No significant difference in NT-ProBNP in either group. Significant decrease in ET-1 after forest bathing compared to controls. No change in ET-1 and the five RAS constituents in the city group compared to baseline. Significant increase in AT2 after forest bathing compared to baseline. No significant difference in controls Significant decrease in serum IL-6 after forest bathing compared to controls. No significant changes in TNF-α or high-sensitive-reactive protein (hCRP) Significant decrease in serum MDA and significant increase in T-SOD after forest bathing compared to controls. No significant difference in controls
Wichrowski et al. (2005) [51]	Two-sample cohort non-randomised	107 participants (42 females) Inpatients on a phase I cardiac rehabilitation programme (does not specify age)	Horticultural therapy 59 participants single session	Patient education class 48 participants single session	Profile of Mood State (POMS) combined POMS Total Mood Disturbance (TDM) Measurements before and after the intervention	Reduced negative POMS scores of “tension”, “depression”, “anger”, “confusion” and “fatigue”, as well as increased “vigour” after horticultural therapy. TMD decreased after horticultural therapy. No changes following patient education class.	Heart rate before and after intervention	Significant decrease in heart rate following horticultural therapy, but no significant change following patient education class

The LTCs reported were chronic obstructive pulmonary disease (COPD; *n* = 1) [47], stroke (*n* = 3) [34,45,46] and CVD (*n* = 9) [48,49,50,51,52,53,54,55,56]. The cardiovascular conditions comprised: high-normal blood pressure (HNBP) or hypertension (*n* = 7) [48,49,52,53,54,55,56], chronic heart failure (*n* = 1) [50], and those included in a cardiopulmonary rehabilitation programme for post cardiac surgery, post myocardial infarction or congestive heart failure (*n* = 1) [51].

The NBIs reported consisted of horticultural therapy (*n* = 1) [51], forest-based interventions (e.g., forest bathing, forest walking, forest therapy, CBT forest therapy or viewing a forest landscape; *n* = 10 [45,47,48,49,50,52,53,54,55,56]) and horse-riding therapy (*n* = 2) [34,46]. Forest-based interventions, such as forest bathing and forest walking, ranged from a single 17 min walk to a 7-day trip. Horse-riding interventions were either 12- or 16-week programmes of weekly sessions, while horticultural therapy consisted of a single 60 min session.

Eleven studies had a control comparison group. Forest-based interventions used an urban or city comparison [45,47,48,49,50,53,55,56] or, in the case of the CBT forest therapy [48], provided control participants with printed educational material only. Horticultural therapy was compared to a patient education class [51]. One horse-riding study compared horse riding and physiotherapy with physiotherapy alone [34]. The studies without a comparison group were one forest study [54] and the qualitative horse-riding study [46]. However, the latter interviewed only participants who took part in the horse-riding arm of a wider randomised control trial (RCT) investigating horse-riding therapy and rhythm and music-based therapy for stroke survivors in late-phase recovery [57,58].

### 3.2. Psychological Outcomes

Twelve studies used quantitative measures to assess the psychological impact of the NBIs. These included the Profile of Mood States (POMS) [59], the Medical Outcomes Study 36-item Short-Form health survey (SF-36) [60], the Semantic Differential (SD) method [61], the Beck Depression Inventory (BDI) [62], the Hamilton Depression Rating Scale (HAM-D17) [63], the Spielberger State-Trait Anxiety Inventory (STAI) [64], and a QoL measurement tool [65] based on the SF-36 and the Duke-UNC Health profile. Other than the SF-36, used only in Beinotti et al.’s [34] single-blind randomised trial by a researcher–rater blind to intervention, all were self-reported measures.

The POMS was used in eight studies, all of which reported improvement following NBIs, although no domain was consistently improved (see Table 4). Decreases in POMS negative domains were reported after a single day of forest therapy in HNBP or hypertension participants [54], as well as when compared to urban or city comparisons in COPD patients [47], HNBP or hypertension participants [49,52,53,56] and chronic heart failure patients [50]. Li et al. [53] also found significant increases in depression in their city comparison group. An increase in the positive domain “vigour” was also observed when compared to urban or city conditions [49,52,53,56]. Horticultural therapy for cardiopulmonary rehabilitation inpatients was reported to decrease negative POMS scores and increase positive “vigour” scores [51]. Additionally, two studies reported significantly reduced total mood disturbance scores following the intervention [51,54].

Beinotti et al. [34] found a significant improvement in functional capacity, physical aspects and mental health factors of the SF-36 in stroke survivors following horse-riding therapy, compared to controls. However, no changes in general health state, vitality or emotional aspects were observed. Measures of “relaxed” and “natural” were increased following forest therapy [54], in addition to “comfortable” scores when compared to an urban condition in participants with HNBP or hypertension [52]. Chun et al. [45] was the only study to measure depression (BDI and HAM-D17) and anxiety (STAI), reporting significant reductions in all three measures from baseline following forest therapy in stroke survivors. One study [51] used a horticultural therapy intervention, while all others used forest therapy interventions.

### 3.3. Physiological Outcomes

Eleven studies included in this review also assessed the physiological impact of NBIs in addition to psychological outcomes. Outcomes measures included blood pressure, heart rate, oxygen saturation, heart rate variability (HRV) and stress hormone secretion, as well as proportions of lymphocytes and pro-inflammatory cytokines. Factors associated with risk for developing COPD [47] and CVD [50,53,56], such as blood cholesterol, C-reactive protein, and NT-ProBNP, were also measured.

Impacts on heart rate (HR) were reported in eight studies. Three forest-based interventions reported significant decreases in HR [52,53,55], while HR of cardiopulmonary rehabilitation patients also decreased significantly following horticultural therapy [51]. Conversely, two studies that measured HR before and after forest walking reported no significant change following the intervention [49,56]. Three studies reported significantly higher parasympathetic activity during forest walking [49,52,55], and one also reported reduced sympathetic activity [49].

Blood pressure was also measured in five studies. Findings showed a significant decrease in blood pressure after forest therapy [54], a significant decrease in diastolic but not systolic blood pressure after forest bathing [49], and a significant decrease in diastolic and systolic blood pressure after forest bathing [56]. However, Li et al. [53] found no significant difference in blood pressure between forest and urban walking conditions, and Sung et al. [48] reported only a marginally significant decrease in systolic blood pressure following viewing forest landscapes. Additionally, a significant increase in oxygen saturation following forest walking compared to city walking was also noted in patients with hypertension [49].

Four studies investigated the impact of forest-based interventions on stress hormone secretion. Serum cortisol and epinephrine significantly decreased in COPD patients after forest bathing but not the city comparison [47]. Similarly, serum cortisol and urinary adrenaline significantly decreased in participants with HNBP after forest therapy [54]. In participants with HNBP or hypertension, there was a non-significant decrease in urinary adrenaline and a significant decrease in urinary dopamine after forest walking compared to urban walking, but both forest and urban walking significantly reduced urinary noradrenaline [53]. Salivary cortisol was also seen to reduce following forest intervention in participants with hypertension [48,54].

Significant reductions of interferon-γ (IFN-γ), interleukin-6 (IL-6) and interleukin-8 (IL-8) were also reported after forest bathing but not the city comparison. In addition, slight decreases in interleukin-1β (IL-1β), tumour necrosis factorα (TNF-α) and C-reactive protein (CRP) were reported after forest bathing but not the city comparison [47]. Mao et al. [50,56] reported a significant decrease in IL-6 but no significant change in TNF-α after forest bathing compared to controls. Mao et al. [50] also reported no significant change in the high-sensitive-reactive protein, but a significant decrease in lipid peroxidation and a significant increase in T-SOD after forest bathing compared to the comparison condition.

As Jia et al. [47] was the only study to investigate the impact of NBIs on patients with COPD, they also assessed the impact of forest bathing on COPD-associated factors. These included pulmonary and activation-regulated chemokine (PARC/CCL-18), surfactant protein D (SP-D) and tissue inhibitor of metalloproteinase (TIMP-1), as well as the proportion of nature killer (NK) cells, nature killer T (NKT) cells and CD8+ T-lymphocytes, all of which are associated with COPD exacerbations. They reported a significant decrease in PARC/CCL-18 and TIMP-1 after forest bathing but not in the city group, and no significant change in SP-D was reported in either group. There was a significant reduction of NK, NKT-like and CD8+ T-cell expression of intracellular perforin after forest bathing but not the city group, and no significant difference in the proportion of NK, NKT-like and CD8+ T-cells, nor their expression of granzyme B.

Two forest-based studies in patients with HNBP or hypertension also assessed specific CVD-related factors. Li et al. [53] found a significant increase in serum adiponectin after forest walking but not the urban comparison; no significant changes in other markers were observed. Mao et al. [56] found a significant decrease in endothelin-1 and homocysteine, RAS constituents including angiotensinogen, angiotensin II type 1 receptor (AT1), angiotensin II type 2 receptor (AT2), and a non-significant reduction in renin and angiotensin II (Ang-II) after forest walking compared to controls. Blood pressure was poorly associated with the change in renin, AT1 and angiotensinogen, while systolic blood pressure was significantly related to Ang-II, endothelin-1 and homocysteine, and diastolic blood pressure was significantly associated with Ang-II and endothelin-1. Similarly, patients with chronic heart failure had a significant decrease in B-type natriuretic peptide (BNP) and endothelin-1 and a significant increase in angiotensin AT2 after forest walking compared to baseline [55]; no significant differences were observed from baseline in the control comparison groups. There was no significant difference in N-terminal pro-BNP in either group.

### 3.4. Qualitative Participant Experiences

The only qualitative study identified in this review explored the experiences of 18 stroke survivors in late-phase recovery following a 12-week group horse-riding programme [46]. Using semi-structured interviews, Pohl et al. [46] explored the impact of the intervention on participants’ physical, psychological and social abilities; their general mood; QoL; and their beliefs regarding the future. Four distinct themes were identified: (1) transformative experiences, encompassing how the intervention altered participants’ view of themselves and their future; (2) human–horse interaction, involving the importance of the physical and emotional relationship with the horse; (3) togetherness and belonging, encompassing the significance of bonding with the group and instructors; (4) the all-in-one solution, describing the richness of interactions with the horses, other group members and staff.

### 3.5. Quality Appraisal

Using the BestBETS quality assessment checklist [39], the studies within this review scored a range of 53–57 (mean = 50.4) out of a possible 64 points (see Appendix A). All but one study [52] scored at least 75%, with four studies scoring over 80% [34,45,48,49]. The quality appraisal facilitated the identification of strengths and weaknesses of the studies. Clear strengths included that all studies stated their aims and objectives, and all had clear protocols reporting intervention and data collection. Seven studies were randomised cohort studies, either having two between-groups comparisons or two cross-over groups. Only two studies reported their randomisation methods [34,45], using an RCT design and randomisation codes to allocate participants to each condition, respectively.

Three out of twelve quantitative studies did not have a control comparison group at baseline [52,54,55]. Song et al. [52,55] did not report a baseline group comparison, although their cross-over design meant all participants completed both the forest and urban sessions. Ochiai et al. [54], however, had no comparison group, limiting the strength of their conclusions on the physiological and psychological effects of forest therapy, although they did collect baseline measures prior to the intervention.

Li et al. [53] used a cross-over design that was not counterbalanced, meaning all participants completed the urban session first followed but the forest session. Another notable confound within this study was a 17 °C temperature difference between the two conditions, with the highest temperature of 37 °C occurring on an urban day.

Additional limitations of the studies include an incomplete description of data, as all but one study [56] reported only *p*-values, and none of the 12 quantitative studies mentioned completing a priori calculations of power or effect size. Sample sizes were generally small, with 11 studies reporting on fewer than 33 participants, often split into two comparison groups so that group sizes did not exceed 23 participants, even with a 2:1 randomisation ratio in favour of the intervention. Only Chun et al. [45] and Wichrowski et al. [51] reported larger sample sizes of 59 and 107, respectively.

## 4. Discussion

The aim of this review was to systematically identify, appraise and synthesise published evidence examining the impact of NBIs on psychological wellbeing of people living with LTCs. Thirteen studies met inclusion criteria; twelve used quantitative methods and one qualitative. All 13 studies reported a significant positive impact of NBIs on a range of psychological wellbeing and physiological outcome measures and appear to support previous research into NBIs’ benefits for physical and mental health, including two recent reviews of forest-based interventions [19,43]. Quality appraisal of the studies showed modest robustness and some methodological weaknesses, although there were four stronger studies [34,45,48,49]. 

Positive psychological impacts of NBIs were particularly demonstrated within the present review as eight of the 13 studies reported improvement on the most frequently used measure, the POMS, following NBI in participants with HNBP or hypertension [47,49,50,51,52,53,54,56], chronic heart failure [50], COPD [47], and those in cardiopulmonary rehabilitation [51]. The most frequently used NBI intervention within this review was a forest-based intervention, although the use of horticultural therapy also demonstrated significant psychological and physiological improvements. The two studies to use horse-riding interventions [34,46] both demonstrated positive psychological benefits, although they did not consider the potential physiological impacts.

Positive physiological impacts of NBIs were also included, notably reporting reduced blood pressure [49,54,56], which highlight NBIs’ potential impact to affect disease progression. Given hypertension’s well-established role in increasing risk of all-cause mortality, cardiovascular and cerebrovascular events [66], and of CKD [67], any mitigation afforded via its reduction can also reduce the risk of stroke [68], CVD events [69] and CVD in individuals with T2DM [70]. 

That the review reveals positive physiological and psychological impacts of NBIs for LTCs suggests parallels with benefits gained from interventions for LTCs, which currently have a more extensive evidence base. Exercise referral programmes, which are often central to self-management of LTCs, also reduce blood pressure in general [71], as well as in individuals with T2DM [72], and can mitigate hypertension, cholesterol and diabetes [73]. A recent systematic review and meta-analysis of RCTs also concluded that exercise interventions improved QoL in T2DM alongside physiological improvements [74], findings previously evidenced for CVD and pulmonary diseases [75]. 

However, this review posits that the benefits of nature exposure through NBIs have the potential to foster multilevel holistic benefits, exceeding that of exercise alone. For example, Pohl et al. [46] identified positive impacts of a horse-riding intervention on participants’ physical, psychological and social abilities, general mood, QoL and future beliefs. Notably, integral to NBIs is relaxation, reducing immediate stressors for the individual and akin to other non-pharmacological interventions, such as massage, decreasing blood pressure and HRV [76]. In this review, both heart rate and HRV were seen to improve following NBIs [49,51,53,55], suggesting a relaxation effect akin to previous forest therapy interventions that do not measure psychological outcomes [77,78]. Evidence for increased relaxation is suggested by reduced stress hormone secretion, such as cortisol [79] and the neurotransmitter dopamine [80]. This was replicated in the forest-based interventions within this review, seeing decreases in cortisol [47,48,54], epinephrine [47], adrenaline [54] and dopamine [53], in line with previous evidence that forest bathing lowers cortisol levels, heart rate and blood pressure in individuals without chronic disease when compared to city environments [77]. Studies within this review also evidenced inflammation reduction, as well as reductions in COPD- and CVD-specific factors following NBIs. Inflammation is linked to increased risk of CVD [81] and COPD [82], while repeated inflammation increase is also associated with the development of depression [83]. Therefore, NBIs appear to have a positive impact over and above the physiological gains.

Evidence for the benefit of NBIs in LTCs is in its infancy, yet recent research comparing forest bathing with compassion mind training in university students found that the interventions had comparable psychological and physiological impacts [84]. This shows promise since although McEwan and colleagues’ study is one of the first to offer forest-based interventions within the UK, it is a much less familiar intervention within the UK in comparison to its routine embedding in public health in Japan and other Asian countries. Efficacy may not have been optimised given the UK’s cultural lack of familiarity with the intervention and by its implementation in winter with cold weather a deterrent to participation and with average air temperatures well below the minimum of 20 °C in forest-based studies in our review.

Indeed, air temperature appears a confound for forest-based interventions within the papers within our review and may moderate their effectiveness, particularly at extremes. Given heat stress is a reported mortality risk factor [85], any higher temperatures may have influenced the significant outcome differences reported, as might lower temperatures since they are associated with higher blood pressure, especially in older adults [86]. Within our review, Li et al. [53] did not observe higher blood pressure in their forest group in line with the significantly lower temperature of that day and therefore concluded that the forest intervention did decrease the blood pressure of participants, although this conclusion should be considered with caution as their study design was not counterbalanced. Similarly, Song et al. [52] also reported temperature difference between the forest and city days, although they had counterbalanced their intervention order. In addition to the likelihood of optimal conditions (including temperature and humidity), there are also individual preferences at work, such that forest bathing on a warm (but not too hot) summer’s day might have more appeal, and hence greater positive impact, than on a cold and wet winter’s day. Such variation in individual temperature preferences and tolerances can depend on regional and cultural norms. 

Whilst showing the positive impacts of NBIs for LTCs, our findings cannot address what works best, for whom and in what context [87]. The “active ingredients” of interventions appear disparate. Horticultural therapy privileges hand–eye coordination, encourages patients into the natural environment and relieves stress, but it also involves social interaction, gentle exercise and gaining satisfaction and a sense of purpose through work and mastery through gaining new skills, well-researched constructs underpinning psychosocial wellbeing. This is also reported following a care-farming programme for people in rehabilitation from mental health problems [88]. Additionally, self-determination theory suggests that those who have a vested interest gain more from the interventions, as seen in exercise referral [89]. Forest-based approaches incorporate more physical movement and exercise or meditation in a forest environment to promote wellbeing but can also involve social interaction. Horse-riding interventions combine both exercise and social aspects, complemented by gaining satisfaction through overcoming a challenge and bond-building with the animal [46]. This suggests that the social aspects of NBIs may be as important for those with LTCs as are the nature elements. For example, CKD is also associated with social isolation that is not solely due to poor mobility [90], and depression symptoms in patients with CKD are associated with increased mortality [91] but can be reduced by building social capital and support. Additionally, a systematic review of reviews and meta-analyses found that peer-support improved clinical, behavioural, and psychological outcomes in patients with T2DM [92], as well as QoL in patients with CKD [93]. Shame and stigma are commonly experienced by people with LTCs, for example, in T2DM [94] and COPD [95]. Shame can emanate from external stigma, e.g., from public or social perceptions, as well as internalised self-stigma [96], making people feel isolated and powerless [97,98]. Perceived stigma is also associated with psychological distress and less social support in T2DM [99] and COPD [100] and negatively impacts medication adherence and help-seeking in COPD [100]. However, by acknowledging vulnerability and reaching out to others, people are able to escape from shame [97].

Nevertheless, psychosocial care of people with LTCs is rarely part of usual condition management, often resulting in higher rates of GP appointments and unplanned primary care admissions [101]. Social prescribing has been introduced in the UK to connect people to emotional and practical support through community groups and statutory services to improve their health and wellbeing [102]. More recently, social prescribing schemes have been piloted by the UK government linking patients with green activities and NBIs [103]. Social prescribing link workers can help improve patient activation in 50+-year-olds with one or more LTC [104]. Social prescribing reduces the number of GP appointments and prescriptions [105], conveying economic benefits along with significant environmental and social co-benefits [106,107]. Individual benefits of social prescriptions include improved self-esteem, psychological wellbeing, sense of empowerment and self-management of LTCs, as well as reduced anxiety, depression and isolation [108].

### 4.1. Clinical Implications

Given that LTCs have burgeoned, the psychological wellbeing of those with LTCs is an important consideration in addition to the physiological impact on morbidity. Many people with LTCs also have a mental health condition [109] or poorer mental health compared to those without an LTC. For example, 40% of people with diabetes report poor psychological wellbeing [110], and COPD is associated with poorer QoL [111]. This review identified improvements in CVD-related factors within populations with HNBP or hypertension, which are precursors for the development of CVD, as well as for stroke and COPD. Providing NBIs alongside or shortly after diagnosis of LTCs may also address multiple factors. For example, the reduction of disease progression and incidence of serious outcomes, including premature mortality, as well as a method of symptom management as they appear to improve psychological coping, which enables people to focus on specific symptom management tasks required for their disease. This is an advantage of NBIs as, since LTCs are defined by their lack of a cure, their use for such individuals would help not only improve their psychological wellbeing but also, in turn, increase their ability to live a fulfilling life that is less marred by their difficult symptoms. As of yet, there are no studies published on the effects of NBIs in patients with diabetes, lung disease, liver disease or CKD, which should be the focus of future research. 

### 4.2. Strengths and Limitations

Studies within this review had some significant vulnerabilities, not least their limited sample sizes and variation in participant culture, race and age. Most research has been conducted in Asia, many studies by the same research group, raising concerns about the generalisability of findings, as well as their cross-cultural relevance and application. Although we sought out grey literature in our searches, publication bias may be evidenced since reporting of significant results or positive effects are more likely to be published than non-significant results or negative effects [112]. Research into NBIs and LTCs is at a germinal stage, which may further increase bias as small studies such as these are also more likely to have positive bias. It is also important to understand the specific benefits of NBIs for more vulnerable groups, particularly ethnic minorities, who have a higher prevalence of LTCs [1], such as CKD [113]. Some argue this is due to the “weathering” hypothesis: the premise that the increased stress experienced throughout the life course as a result of structural racism leads to poorer outcomes [114]. 

Additionally, the validity of some of the studies is in question, particularly in relation to the constraints placed on participants. Of note was that a number of the forest-based studies did not permit participants to consume alcohol or caffeine or to smoke during the interventions [49,50,53,55,56]. Whilst this may control for potential confounds, the impacts of abstinence are unexamined. Notably, Ochiai et al. [54] did not permit participant interaction during the programme or use of mobile phones. With no comparison group, the impact they reported could be attributed to the restricted social interaction, either positively or negatively. 

### 4.3. Review Limitations

Limitations of the present review include the relatively small number of studies included, which constrain comparison or firm conclusions. In addition, the majority of studies reported *p*-values only; therefore, it was not possible to conduct a meta-analysis. Nevertheless, a meta-analytic approach to compiling the effects of NBIs on LTCs is a direction for future research that we would recommend. Our review included English language papers only, yet elicited studies were conducted predominantly in Asia, in which English is not a first language, which may have unintentionally excluded further studies.

The LTCs investigated were circumscribed and did not cover the widest range of debilitating conditions. However, in constraining our focus to CVD, stroke, lung disease, liver disease, CKD and T2DM, we were mindful of their population prevalence, morbidity and mortality, and their impact on healthcare demand and patient QoL, as well as their amenability to lifestyle and behavioural interventions for condition management [74,75]. 

Our review also highlighted absences in the literature thus far, notably the potential barriers to implementing NBIs to enable their integration into mainstream healthcare practice. Such barriers may include people being deterred from participation by aversions to or fear of animals, the presence of allergies and access to transport. Depending on the intervention, physical ability can be a barrier for many NBIs, as people with mobility disabilities are less able to access greenspaces, even when they have the desire to [115]. Similarly, motivation or willingness to engage is likely to vary in relation to effort, for example, a seven-day forest bathing trip compared to a 60-minute horticultural therapy session in a hospital garden. This is where social capital, as mentioned above, could also increase motivation. Individual differences in nature connectedness may also moderate the impact of NBIs as those with more connection to nature may be more likely to get greater benefits from such interventions [116]. This could be a possible moderator of individual benefit from NBIs, assuming that greater nature connectedness would mean greater positive impact, as those with a greater affinity with nature are more likely to gain benefit from it. 

Further limitations of the present review include the inability to answer the question of “what NBIs work best for which LTC in what contexts”. This is largely due to the limited number of studies covering a range of both LTCs and NBIs, but this gap in the evidence warrants further research and this limitation is not constrained to the present review. For example, Wilkie and Davinson [117] conducted a scoping review of the prevalence and effectiveness of NBIs on health-related behaviours and outcomes, reporting that there is little evidence for the long-term efficacy of NBIs. Effects were small but positive and were assessed over a short period of time with no follow-up. These are also limitations of the studies included in the present review. Future research should investigate the strength of these effects over time. Wilkie and Davinson also suggest investigating if different delivery lengths of NBIs have an impact on dose–response and thus inform treatment plans for “minimum duration for maximum benefit” (p. 7).

Additionally, it may be that different durations of NBIs may work better for those with certain LTCs than for others. Similarly, different delivery methods may suit different patient groups more so than others. For instance, virtual reality is a recent phenomenon in NBIs and evidence suggests delivering virtual NBIs demonstrates similar effects on psychological wellbeing to that of physical environments [118,119]. However, the caveat is a need for further understanding of which components of these virtual and physical environments are interacting with nature to achieve positive impacts on psychological wellbeing. Nevertheless, the use of virtual environments within NBIs may address some of the barriers to engagement discussed above, particularly for those who are unable to access real-life nature environments. For example, a recent study that was interrupted by the COVID-19 restrictions also found that conducting many elements of NBIs remotely also had therapeutic gain for participants with depression [120].

## 5. Conclusions

The area of NBIs for LTCs shows limited but positive effects on a range of physical health and psychological wellbeing outcomes, consistent with similar studies demonstrating the benefits of NBIs for health and wellbeing in other populations. Additional support is provided by qualitative evidence that highlights the potentially transformative experiences of participants. Future research could explore NBIs for other LTCs and incorporate additional factors that may moderate effectiveness, such as nature connectedness.

## Figures and Tables

**Figure 1 ijerph-19-03214-f001:**
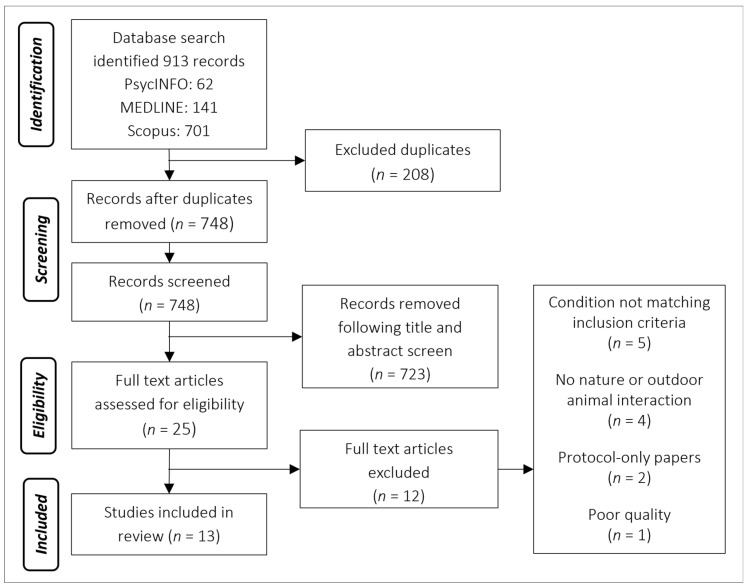
PRISMA flow diagram depicting the process of screening.

**Table 1 ijerph-19-03214-t001:** PICO framework.

PICO	
P—Population	People with long-term physical health conditions limited to cardiovascular disease, stroke, lung and liver disease, type II diabetes and chronic kidney disease
I—Intervention	Nature-, green- or outdoor animal-based interventions
C—Comparison	Treatment as usual, urban environments or no comparison
O—Outcome	Evaluation of effectiveness, as measured by improvements in psychological wellbeing and/or quality of life (QoL)

**Table 2 ijerph-19-03214-t002:** Free text search terms for nature- and animal-based interventions with long-term conditions. These were combined with the Boolean operator AND.

Search Terms	
Nature-based interventions	((garden* OR green OR horticultur* OR “nature-based” OR “nature based”) N2 (therap* OR intervention* OR proximity)) OR ((healing OR restorative OR wander) N2 garden) OR “green prescri*” OR “social prescri*” OR “nature prescri*” OR “nature play” OR “park prescri*” OR “garden prescri*” OR “green space*” OR greenspace* OR “green exercise” OR “green infrastructure” OR “community garden*” OR “community allotment*” OR allotment* OR “outdoor exercise” OR “blue space*” OR “blue gym*” OR “green gym*” OR “park prescri*” OR “eco therapy” OR “eco-therapy” OR “wilderness therapy” OR “wilderness-therapy” OR “care-farming” OR “care farming” OR “farm therapy” OR “farm-therapy” OR “forest bathing” OR “forest-bathing” OR “environmental volunteering” OR “wild play” OR “nature play” OR “animal assisted therap*” or “animal-assisted therap*” OR “animal therap*” OR “pet therap*” or “pet-assisted therapy” OR “equine assisted therap*” OR “equine-assisted therap*” OR “canine assisted therap*” OR “canine-assisted therap*”
Long-term conditions	Cardiovascular or “cardiovascular disease” or Hypertension or “high blood pressure” or “Coronary Heart Disease” or “heart disease” or CHD or “Coronary Disease” or “vascular disease” or “Heart failure” or “Pulmonary Heart Disease” OR “Pulmonary disease” or “Respiratory disease” or Asthma or “Chronic Obstructive Pulmonary Disease” or COPD OR “Liver disease” or “Chronic liver disease” or “liver cirrhosis” or “Fatty Liver” or Hepatitis or “hepatic disease” OR “type II diabetes” or “type two diabetes” or “type 2 diabetes” or Diabetes or T2DM or “diabetes mellitus” OR “Kidney disease” or “Chronic kidney disease” or CKD or “renal insufficiency” or “chronic renal insufficiency” or “renal disease” or “chronic renal disease” or “kidney failure” or “renal failure” or AKI or “acute kidney injury”

* Truncation symbol which when used at the end of search terms finds any string of characters in that position; for example, therap* would identify therapist, therapies, therapy etc.

**Table 4 ijerph-19-03214-t004:** Summary of POMS positive outcomes after the intervention (decreased negative, increased positive) by domain. + denotes this domain was significantly improved following the intervention.

	Negative					Positive
Author (Date) [Ref]	Tension-Anxiety	Depression	Anger-Hostility	Fatigue	Confusion	Vigour
Jia et al. (2016) [47]	+	+	+			
Song et al. (2015) [52]	+	+	+	+	+	+
Li et al. (2016) [53]		+	+	+	+	+
Ochiai et al. (2015) [54]	+		+		+	
Wu et al. (2020) [49]	+	+		+	+	+
Mao et al. (2012) [56]		+	+	+	+	+
Mao et al. (2017) [50]	+	+	+		+	
Wichrowski et al. (2005) [51]	+	+	+	+	+	+

## Data Availability

Not applicable.

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
