# Peer review of "Nature-Based Interventions for Psychological Wellbeing in Long-Term Conditions: A Systematic Review"

_ijerph, 2022, doi:10.3390/ijerph19063214_

Round 1

Reviewer 1 Report

The manuscript presents a high-quality piece of research reviewing the literature on the impact of nature-based interventions on patients with long-term conditions. I am happy with the topic, the methodology, and the results. 

I have only two little questions that may translate into recommendations:

  1. First, I couldn't find any references to the duration of the positive physiological and psychological impacts of NBIs for LTCs (I hope I didn't miss them): while they are undoubtedly there, how long-lasting are these effects? This is an important concern for any intervention. 
  2. Related to the limitations of the qualitative path you took in your systematic review, can you suggest a quantitative study design to be carried out in the future to better document the effects of NBIs on LTCs patients?

Author Response

Response to Reviewer 1 Comments

Point 1: First, I couldn't find any references to the duration of the positive physiological and psychological impacts of NBIs for LTCs (I hope I didn't miss them): while they are undoubtedly there, how long-lasting are these effects? This is an important concern for any intervention.

Response 1: Thank you for pointing out this valuable suggestion to us. We have added the below text to lines 602-613 as follows:

“Further limitations of the present review include the inability to answer the question of “what NBIs work best for which LTC in what contexts.” This is largely due to the limited number of studies covering a range of both LTCs and NBIs but this gap in the evidence warrants further research and this limitation is not constrained to the present review. For example, Wilkie & Davinson [118] conducted a scoping review of the prevalence and effectiveness of NBIs on health-related behaviours and outcomes, reporting that there is little evidence for the long-term efficacy of NBIs. Effects were small but positive and were assessed over a short period of time with no follow-up. These are also limitations of the studies included in the present review. Future research should investigate the strength of these effects over time. Wilkie & Davinson also suggest investigating if different delivery lengths of NBIs have an impact on dose-response, and thus inform treatment plans for “minimum duration for maximum benefit” (pp. 7).”

Point 2: Related to the limitations of the qualitative path you took in your systematic review, can you suggest a quantitative study design to be carried out in the future to better document the effects of NBIs on LTCs patients?

Response 2: We have included a reflection on why meta-analysis was not possible for the present review. The below text has been added to lines 577-580 as follows:

“In addition, the majority of studies reported p-values only, therefore it was not possible to conduct a meta-analysis. Nevertheless, a meta-analytic approach to compiling the effects of NBIs on LTCs is a direction for future research that we would recommend.”

Reviewer 2 Report

The manuscript in evaluation reports the results of a review exploring the effect of NBIs on LTC patients. The systematic search through online databases led to a set of 13 studies that overall supported the use of NBIs for these patients. Although limited in the number of studies reported (not to the fault of the authors!), the article is well organized and the results are clearly reported.  This manuscript is really interesting and adds to the existent literature a valuable contribution.

While I appreciated the clear format of the presentation, I would have enjoyed also seeing a meta-analysis of the summarized effects of NBIs on well-being. In fact, the physiological measures reported were really different between the different papers reviewed, however, the psychological measures could be summed up and analyzed through a meta-analytic process. I understand that this goes beyond the scope of the manuscript in evaluation, but I hope the authors would take it as a suggestion for future work on the theme. 
Another point that could be taken into consideration, maybe also to be integrated with the present discussion, is the effect of nature's immersive experience with possible virtual environment applications. Nowadays, virtual reality allows people to experience realistic scenarios such as a forest walk or mountain climbing. For example, Yu et al (2018) and Mattila et al. (2020) both reported increased psychological well-being after exposition to a virtual forest environment. This could be a huge leap forward for the treatment of LTC patients, as it could ensure the accessibility to natural environments even for people living far from the natural environment, e.g., in a big city or industrial landscape. Please take into consideration this suggestion to increase the quality of the discussion.

Yu, C. P., Lee, H. Y., & Luo, X. Y. (2018). The effect of virtual reality forest and urban environments on physiological and psychological responses. Urban forestry & urban greening35, 106-114.   Mattila, O., Korhonen, A., Pöyry, E., Hauru, K., Holopainen, J., & Parvinen, P. (2020). Restoration in a virtual reality forest environment. Computers in Human Behavior107, 106295.
I am supportive of the publication of this paper in its actual form.

Author Response

Response to Reviewer 2 Comments

Point 1: While I appreciated the clear format of the presentation, I would have enjoyed also seeing a meta-analysis of the summarized effects of NBIs on well-being. In fact, the physiological measures reported were really different between the different papers reviewed, however, the psychological measures could be summed up and analyzed through a meta-analytic process. I understand that this goes beyond the scope of the manuscript in evaluation, but I hope the authors would take it as a suggestion for future work on the theme.

 Response 1: We have included a reflection on why meta-analysis was not possible for the present review. The below text has been added to lines 577-580 as follows:

“In addition, the majority of studies reported p-values only, therefore it was not possible to conduct a meta-analysis. Nevertheless, a meta-analytic approach to compiling the effects of NBIs on LTCs is a direction for future research that we would recommend.”

Point 2: Another point that could be taken into consideration, maybe also to be integrated with the present discussion, is the effect of nature's immersive experience with possible virtual environment applications. Nowadays, virtual reality allows people to experience realistic scenarios such as a forest walk or mountain climbing. For example, Yu et al (2018) and Mattila et al. (2020) both reported increased psychological well-being after exposition to a virtual forest environment. This could be a huge leap forward for the treatment of LTC patients, as it could ensure the accessibility to natural environments even for people living far from the natural environment, e.g., in a big city or industrial landscape. Please take into consideration this suggestion to increase the quality of the discussion.

Response 2: We have combined this valuable point with suggestions from other reviewers and added the below text to lines 614-625 as follows:

“Additionally, it may be that different durations of NBIs may work better for those with certain LTCs than for others. Similarly, different delivery methods may suit different patient groups more so than others. For instance, virtual reality is a recent phenomenon in NBIs and evidence suggests delivering virtual NBIs demonstrates similar effects on psychological wellbeing to that of physical environments [119, 120]. However, the caveat is a need for further understanding of which components of these virtual and physical environments are interacting with nature to achieve the positive impacts on psychological wellbeing. Nevertheless, use of virtual environments within NBIs may address some of the barriers to engagement discussed above, particularly for those who are unable to access real-life nature environments. For example, a recent study that was interrupted by the COVID-19 restrictions also found that conducting many elements of NBIs remotely also had therapeutic gain for participants with depression [121].”

Reviewer 3 Report

Summary: The authors report their examination of 13 studies of the effects of NBI’s for people with LTC’s showing that there were several positive effects on both psychological and physiological variables.

General: 

This report is a valuable contribution supporting the goal of determining the role of NBI’s in helping people with various challenging conditions. Identifying relevant studies, detailing their methods, and summarizing their findings is useful for researchers who wish to take next steps in this important area.  I appreciated the well-organized and well-written report.

One of the major strengths of the manuscript is highlighting the limitations of the existing research.  This is the part of the paper that I believe can be strengthened.  I would suggest the authors offer a more systematic identification of limitations, what is not known after the analysis of the 13 studies, and provide suggestions for the future of the field in applying NBI’s to LTC’s. 

The selection process for the studies was well-described. It may be useful for the authors to characterize in more detail the 12 full-text studies that were excluded so the reader has confidence in why they were dropped.

Because of the prevalence of meta-analyses in research that attempts to synthesize multiple studies, the authors should describe why they did not use this technique, so that readers are not distracted by wondering.

Specific:

(111-113)  Confusing:  “This review considered NBIs conducted outside or by bringing nature inside but was limited to animal-based interventions conducted outside, not including, for example, pet-assisted therapy conducted at a hospital bedside”. Seems to be saying that only outside animal-based interventions were included.

Overall, this is a nice contribution.

Author Response

Response to Reviewer 3 Comments

Point 1: One of the major strengths of the manuscript is highlighting the limitations of the existing research. This is the part of the paper that I believe can be strengthened. I would suggest the authors offer a more systematic identification of limitations, what is not known after the analysis of the 13 studies, and provide suggestions for the future of the field in applying NBI’s to LTC’s.

Response 1: We have taken on board your advice to strengthen the limitations section and combined this with suggestions from other reviewers by discussing what has not been answered by our review. We have added the below text to lines 602-625 as follows:

“Further limitations of the present review include the inability to answer the question of “what NBIs work best for which LTC in what contexts.” This is largely due to the limited number of studies covering a range of both LTCs and NBIs but this gap in the evidence warrants further research and this limitation is not constrained to the present re-view. For example, Wilkie & Davinson [118] conducted a scoping review of the prevalence and effectiveness of NBIs on health-related behaviours and outcomes, reporting that there is little evidence for the long-term efficacy of NBIs. Effects were small but positive and were assessed over a short period of time with no follow-up. These are also limitations of the studies included in the present review. Future research should investigate the strength of these effects over time. Wilkie & Davinson also suggest investigating if different delivery lengths of NBIs have an impact on dose-response, and thus inform treatment plans for “minimum duration for maximum benefit” (pp. 7).

“Additionally, it may be that different durations of NBIs may work better for those with certain LTCs than for others. Similarly, different delivery methods may suit different patient groups more so than others. For instance, virtual reality is a recent phenomenon  in NBIs and evidence suggests delivering virtual NBIs demonstrates similar effects on psychological wellbeing to that of physical environments [119, 120]. However, the caveat is a need for further understanding of which components of these virtual and physical environments are interacting with nature to achieve the positive impacts on psychological wellbeing. Nevertheless, use of virtual environments within NBIs may address some of the barriers to engagement discussed above, particularly for those who are unable to ac-cess real-life nature environments. For example, a recent study that was interrupted by the COVID-19 restrictions also found that conducting many elements of NBIs remotely also had therapeutic gain for participants with depression [121].”

Point 2: The selection process for the studies was well-described. It may be useful for the authors to characterize in more detail the 12 full-text studies that were excluded so the reader has confidence in why they were dropped.

Response 2: We have added the below text to lines 150-154 as follows:

“The 12 full-text articles that were excluded prior to analysis were deemed not appropriate for inclusion. Reasons for exclusions included: not having a condition that matched the inclusion criteria for an LTC, (n=5), not containing a nature or green element (n=4), protocol-only papers and did not contain any data (n=2), deemed poor quality as it did not report any data (n=1).”

Point 3: Because of the prevalence of meta-analyses in research that attempts to synthesize multiple studies, the authors should describe why they did not use this technique, so that readers are not distracted by wondering.

Response 3: We have included a reflection on why meta-analysis was not possible for the present review. The below text has been added to lines 577-580 as follows:

“In addition, the majority of studies reported p-values only, therefore it was not possible to conduct a meta-analysis. Nevertheless, a meta-analytic approach to compiling the effects of NBIs on LTCs is a direction for future research that we would recommend.”

Point 4: (111-113) Confusing: “This review considered NBIs conducted outside or by bringing nature inside but was limited to animal-based interventions conducted outside, not including, for example, pet-assisted therapy conducted at a hospital bedside”. Seems to be saying that only outside animal-based interventions were included.

Response 4: You have understood correctly that only outside animal-based interventions were included in the present review. Nevertheless, we have reworded lines 111-118 to clarify this statement as follows:

“This review considered NBIs if the interventions contained a nature or green element, whether conducted inside or outside. Animal-based interventions conducted inside, for example, pet-assisted therapy conducted at a hospital bedside, were not included in this review as they did not meet the inclusion criteria of containing a nature or green element. However, those delivered within an outdoor natural environment were included.”